# Retrospective Review of Intra-Cerebrospinal Fluid (CSF) Drug Delivery in CNS Malignancies: Safety, Clinical Efficacy and Pharmacokinetic Profiles of Intracerebroventricular (ICV), Lumbar Intrathecal (LIT), and Intra-Cisterna Magna (ICM) Injections

**DOI:** 10.3390/cancers17081263

**Published:** 2025-04-09

**Authors:** Grace Y. Lee, Marcie A. Glicksman, Rajan Patel, Saaz Malhotra, Nathan Moelis, Nisheka N. Vanjani, Priya Kumthekar

**Affiliations:** 1Feinberg School of Medicine, Northwestern University Feinberg, Chicago, IL 60611, USA; grace.lee2@northwestern.edu (G.Y.L.); saaz.malhotra@northwestern.edu (S.M.); nisheka.vanjani@northwestern.edu (N.N.V.); 2EnClear Therapies, Newburyport, MA 01950, USA; m.glicksman@encleartherapies.com (M.A.G.); r.patel@encleartherapies.com (R.P.); moelis.n@northeastern.edu (N.M.); 3Department of Neurology, Northwestern Memorial Hospital, Chicago, IL 60611, USA

**Keywords:** cerebrospinal fluid (CSF), blood brain–barrier (BBB), intrathecal (IT) chemotherapy, lumbar puncture (LP), intracerebroventricular (ICV), Ommaya reservoir

## Abstract

Intra-cerebrospinal fluid (CSF) drug delivery is being increasingly used to target CNS malignancies. There are multiple methods of intra-CSF delivery, each with its unique challenges and benefits. To the authors’ knowledge, there are no studies that compare intra-CSF drug delivery methods head to head. The aim of this retrospective review was to review and compare the safety, efficacy, and pharmacokinetic profiles of three intra-CSF delivery methods: intracerebroventricular (ICV), lumbar intrathecal (LIT), and intra-cisterna magna (ICM). We find that the safety profiles of both ICV and LIT injections show mostly mild to moderate procedure-associated AEs and that ICV delivery achieves therapeutic goals more consistently than the other intra-CSF delivery methods. There are insufficient data to show dose-related response with intra-CSF delivery. These findings highlight the importance and need for development of novel tools to improve upon intra-CSF delivery.

## 1. Introduction

The blood–brain barrier (BBB) and blood–cerebrospinal fluid barrier (BCSFB) are highly selective physiological barriers to the central nervous system (CNS) which separate blood vasculature from neural tissues. The barriers contain an elaborate network of tight junctions and various efflux transporters, which prevent the free diffusion of macromolecules into the brain parenchyma [1,2]. For molecules to freely cross the BBB/BCSFB, they must be highly lipid-soluble and have a molecular weight less than 500 Da [3]. While this selectivity maintains a protective barrier around the CNS, the same barrier poses a challenge to the delivery of many therapeutics designed to target CNS disease, such as CNS malignancies. Despite the continuously increasing global incidence of CNS malignancies, the treatment of brain tumors and metastases lags far behind that of peripheral malignancies, owing in part to the challenges in delivery of the drug through the BBB/BCSFB [4,5,6].

Circumventing the challenges in systemic delivery relying on vasculature, local delivery modalities are increasingly utilized to target CNS diseases. One such modality includes injection directly into the cerebrospinal fluid (CSF) (i.e., intra-CSF delivery). There are three main access points for intra-CSF injections: intracerebroventricular (ICV), lumbar intrathecal (LIT), and intra-cisterna magna (ICM). The characteristics of each method are summarized in Table 1. ICV drug delivery requires the surgical implantation of a ventricular access device (e.g., Ommaya reservoir) in the subgaleal space under the scalp, with a catheter to access the lateral ventricles [7], and ICM drug delivery also involves surgical introduction of a catheter to facilitate injection into the subarachnoid space between the cerebellum and medulla oblongata [8]. LIT drug delivery is performed through a lumbar puncture, in which a sterile needle is placed into the cerebrospinal space between two vertebrae. The decision as to which CNS delivery location is most suitable should be made based on the targeted area(s). For instance, the lumbar IT approach is preferred for spinal cord injury as it enables high local concentrations in the spinal cord. In contrast, one might choose ICM to achieve higher concentrations, especially in the brain stem region, but the feasibility of this route in humans is questionable. The cisterna magna is a large CSF compartment and is frequently used in studies with non-human primates and other animals. However, it is infrequently used as an access point in humans mainly because inserting a needle near the vital centers of the medulla oblongata could cause brain stem injury.

While intra-CSF drug delivery provides the benefit of bypassing the BBB/BCSFB and providing direct access to the CNS, it is not without its unique challenges. The rapid turnover and flow of the CSF often limit CNS exposure to therapeutics. In humans, the CSF production rate is estimated to be between 0.3–0.6 mL/min, resulting in a turnover of around five times per day (for a total CSF volume of 150 mL) [16]. This rapid turnover has diluting effects on drug levels, evidenced in their delay in reaching maximal concentration and the exponential decay in their concentration following dosage [17]. Moreover, intra-CSF delivery requires invasive procedures to secure CSF access and to implant the necessary devices, leaving potential for both procedure- and device-related complications and adverse events (AEs).

Given the unique access points of each of the three delivery methods, each present with distinct strengths and challenges. This study aims to review and compare the safety, pharmacokinetic, and clinical efficacy profiles of the different intra-CSF drug delivery methods in the treatment of CNS malignancies.

## 2. Materials and Methods

A retrospective literature search was conducted using PubMed to identify clinical trials published from 2000 to 2024 that used ICV, LIT, and/or ICM as a route of drug administration. The search was conducted with the following keywords: “(intracerebroventricular OR ICV OR intraventricular) AND (drug delivery OR drug administration) AND (CSF)” for ICV; “(intrathecal OR IT) AND (drug delivery OR drug administration) AND (CSF)” for LIT; and “(cisterna magna OR intra cisterna magna OR ICM OR IT-CM) AND (drug delivery OR drug administration) AND (CSF)” for ICM (Figure 1). The records identified through this primary search were screened by two independent reviewers for clinical studies specifically addressing neoplastic CNS disease, defined as primary and/or metastatic tumors of the CNS. Preclinical studies, as well as studies focusing solely on properties of the drug rather than the delivery methods, were excluded. Then, the reviewers assessed full-text articles for eligibility of analysis; papers containing outcomes of interest for safety, clinical efficacy, and pharmacokinetics were identified, categorized, and included for analysis.

## 3. Results

The primary search yielded 38 ICV, 110 LIT, and six ICM publications addressing a range of disease states (Table 2 and Table 3). When narrowed down to CNS neoplastic disease, there were 19 ICV, 22 LIT, and zero ICM publications. Given our focus on the safety, pharmacokinetic, and clinical efficacy profiles of each delivery method, we excluded studies not containing outcomes of interest. We defined outcomes of interest as delivery-specific AEs for safety, response and/or survival for clinical efficacy, and dosimetry and/or elimination data for pharmacokinetics. We also excluded studies in which patients received therapeutics through multiple intra-CSF access points (e.g., both Ommaya reservoir and lumbar puncture). This left a total of 11 ICV and one LIT publications for safety analysis, 12 ICV and one LIT publications for clinical efficacy analysis, and four ICV publications for pharmacokinetics analysis (Figure 1). These studies encompassed secondary CNS cancers as well as primary brain malignancies including ependymoma, medulloblastoma, and primary CNS lymphoma (Table 2).

### 3.1. Intracerebroventricular Findings

#### 3.1.1. ICV Safety

A total of 11 ICV studies reported on safety outcomes, as summarized in Table 3. Overall, the most frequently reported AEs were headache, nausea, and vomiting. These are common AEs associated with chemotherapy drugs. In the reports, there was not a distinction between AEs associated with the drug versus those related to the ICV delivery modality. Most often, these AEs were low-grade and transient, and patients were able to resume treatment after resolution (Table 4). Of note, infection was a less frequently reported but more clinically severe complication, resulting in the discontinuation of treatment for patients in Sandberg et al. and Blaney et al. (2013)’s publications [21,25]. Arachnoiditis, a feared complication of ICV device implantation, was reported directly as an AE in Kumthekar et al. (2022) and Blaney et al. (2013)’s studies [19,25], with frequencies noted below in Table 3. Blaney et al. (2013) [25] defined arachnoiditis through the presence of fever, nausea and/or vomiting, and headache and/or back pain. While the other studies did not directly identify arachnoiditis as a complication, the most frequently reported AEs seem to correlate with the constellation of arachnoiditis symptoms as defined by Blaney et al. (2013) [25].

#### 3.1.2. ICV Clinical Efficacy

All 12 ICV publications reported on clinical efficacy in the form of response and/or survival, as summarized in Table 5. The response rate was affected by route of delivery but also related to the therapeutic used and other aspects of the patients’ clinical course at treatment. These diseases were heterogeneous in outcome as median survival varied greatly between leptomeningeal disease versus primary brain tumors (i.e., medulloblastoma, ependymoma) and parenchymal brain metastases, but are nonetheless provided here for completion along with response rates.

#### 3.1.3. ICV Pharmacokinetics

There were four publications that specifically examined the pharmacokinetics of drugs after ICV administration, and they are summarized in Table 6. These studies were performed with different therapeutics. Blaney et al. (2013) [25] focused on intraventricular topotecan in children with neoplastic meningitis and were able to consistently exceed the therapeutic target concentration of 1 ng/mL. The mean CSF concentration-time profile indicated a sustained exposure, with simulations predicting that over 99.9% of patients would achieve the target concentration threshold. Fleischhack et al. [29] examined the feasibility of intraventricular etoposide in patients with metastatic brain tumors. CSF peak levels exceeded systemic levels by more than 100-fold. Notably, there was significant interindividual variability.

Rubenstein et al. [24] investigated the use of intraventricular immunochemotherapy with rituximab and methotrexate in patients with recurrent CNS lymphoma. Notably, the elimination rate of rituximab was significantly slower when co-administered with methotrexate (0.36/h) compared to rituximab monotherapy (0.84/h), indicating prolonged CSF retention. Additionally, serum rituximab concentrations increased gradually over the course of treatment, reflecting slow systemic absorption from the CSF.

The pharmacokinetics of ICV administration was explored by Kumthekar et al. (2022) [19], whose study showed that ICV trastuzumab delivery was characterized by rapid distribution within the CSF and limited systemic absorption. Systemic absorption was not directly quantified, but the authors note that serum trastuzumab concentrations were always measurable before administration of doses subsequent to the first and that there were slight accumulations in trastuzumab concentrations with each dose, reflecting overall minimal, delayed systemic absorption and slow transfer from the CSF to the bloodstream.

### 3.2. Lumbar Intrathecal Findings

#### 3.2.1. LIT Safety

Only one study by Fan et al. [30] reported data on safety outcomes related to LIT delivery. AEs determined to be possibly related or related to drug delivery spanned Common Terminology Criteria for Adverse Events (CTCAE) grades 1 to 4. Most frequently, the AEs were low-grade and transient. There was one reported case of grade 4 myelosuppression (Table 7).

#### 3.2.2. LIT Clinical Efficacy

Glantz et al. (2010) [26] provide further insights into the efficacy of different IT administration routes. When PFS was compared between LIT and ICV administration of sustained-release cytarabine, there was no significant difference observed (29 days vs. 43 days, *p* = 0.35). However, a notable distinction was found in patients treated with methotrexate. ICV administration of methotrexate resulted in a statistically significant improvement in PFS compared to LIT administration (43 days vs. 19 days, *p* = 0.048).

## 4. Discussion

Intra-CSF delivery of therapeutics has been a decades-old strategy to overcome the BBB/BCSFB and facilitate direct access to the CNS, while reducing peripheral exposure. This approach may hold particular promise in the treatment of CNS neoplasms, where traditional systemic therapies are often limited by the BBB and peripheral toxicity. Circumventing the challenges in systemic delivery relying on vasculature, local delivery modalities are increasingly utilized to target CNS diseases. One such modality includes injection directly into the CSF (i.e., intra-CSF delivery). Over the years, various intra-CSF delivery methods such as ICV, LIT, and ICM have been explored to enhance drug distribution within the CSF and target tumor cells more effectively. While these strategies have shown potential in specific clinical settings, each delivery method carries unique advantages, limitations, and risks that influence their clinical applicability. Understanding these factors is essential for optimizing treatment protocols and improving outcomes for patients with CNS neoplasms, especially considering the evolving therapeutic options and novel therapeutics including immunotherapy, cellular therapies, and targeted therapies [31]. This includes a better understanding of the pharmacokinetics when chemotherapy is administered directly into the CSF. It is not possible to generalize, as the characteristics of the drug can influence the characteristics of the uptake and clearance from the CSF. The patient population may also influence the pharmacokinetics. This was noted in some of the studies [29].

ICV and LIT are the two most common CSF delivery options. ICM delivery is not typically a first-line treatment for CNS neoplastic diseases in humans due to several significant limitations. While it offers the potential for direct drug delivery to the CSF, the procedure is invasive and carries inherent risks, including infection, bleeding, and neurological injury. Additionally, ICM delivery is technically challenging and can be difficult to perform in patients with anatomical variations or conditions that complicate access. The development of more targeted and minimally invasive alternatives, such as intrathecal or intraventricular drug delivery, has largely supplanted ICM for targeting of CNS neoplasms. This trend was directly reflected in the results of our review, with the literature search yielding minimal results of studies in which ICM was used as the main modality of drug delivery.

From a safety perspective, the papers cited here indicate a low risk from intra-CSF delivery. There were few adverse events thought to be directly related to the intrathecal drug delivery itself and as a whole appear to be similarly, if not better, tolerated than systemically administered drugs, particularly with regards to the paucity of systemic related toxicity when delivered directly to the CNS. Despite the safety seen, the use of CSF direct delivery has been limited, perhaps due to the limited improvement seen in patient outcome measures. There is a fine balance between CSF production (0.3–0.6 mL/min), volume (150 mL), and removal/turnover (every 5 h) [16]. The high turnover of CSF and the active efflux makes even direct administration to CSF particularly challenging for drug distribution, retention, and parenchymal penetration.

Novel tools are being developed to improve upon local or intra-CSF drug delivery that will ideally lead to improved patient outcomes in the near future. For example, convection-enhanced delivery (CED) is a method of targeted, local drug delivery to the CNS, utilizing pressure for delivery with catheters inserted directly into the target region. One CSF delivery strategy has utilized a system of continuous IT drug delivery that exchanges fluid between the lateral ventricles and showed a 14% complication rate; efficacy data are pending [32]. Another strategy is called ventriculolumbar perfusion (VLP) chemotherapy, where the drug is administered continuously by a pump and with a simultaneous lumbar drainage to relieve the hydrostatic pressure produced. However, VLP chemotherapy has shown only a similar or slightly better response rate when compared to conventional intra-CSF chemotherapy, but with moderate to severe side effects [33]. Another strategy to overcome these limitations is the development of a novel CSF delivery tool, the EnTrega system. This system is comprised of two fluid access points, an intraventricular and an intralumbar access point, with an external motor to move CSF in a closed loop. In addition, there is an external and disposable pressure and flow sensor array, providing real-time pressure sensor data from both access points, as well as warning alarms for temperature, bubble, and pressure outside of the safety thresholds. This system has shown improved CNS drug concentrations by using this CSF recirculation method, and has also shown for the first time, to the authors’ knowledge, the presence of brain parenchymal drug uptake with CSF delivery [34,35].

Overall, intra-CSF drug delivery has been a strategy used for many years in the setting of CNS neoplastic disease with an acceptable safety profile, albeit with limited efficacy data. Novel strategies to optimize intra-CSF delivery may help to more effectively improve outcomes for patients with cancer and other neurologic diseases. As our understanding of the complexities of the blood–brain and blood–cerebrospinal fluid barriers continues to evolve, there is growing potential for developing more precise, targeted delivery systems that can maximize drug bioavailability while minimizing systemic side effects. Advances in technologies, such as nanocarriers, genetically engineered vectors, and implantable pumps, hold promise for enhancing the efficiency and safety of intra-CSF drug administration, particularly for patients with CNS neoplasms. However, challenges related to procedural invasiveness, patient variability, and potential complications must be addressed in parallel to ensure that these therapies can be safely and widely applied. Ultimately, with continued research and clinical innovation, optimized intra-CSF delivery may offer a transformative approach to treating CNS diseases, providing more effective and personalized care for patients with these conditions.

## Figures and Tables

**Figure 1 cancers-17-01263-f001:**
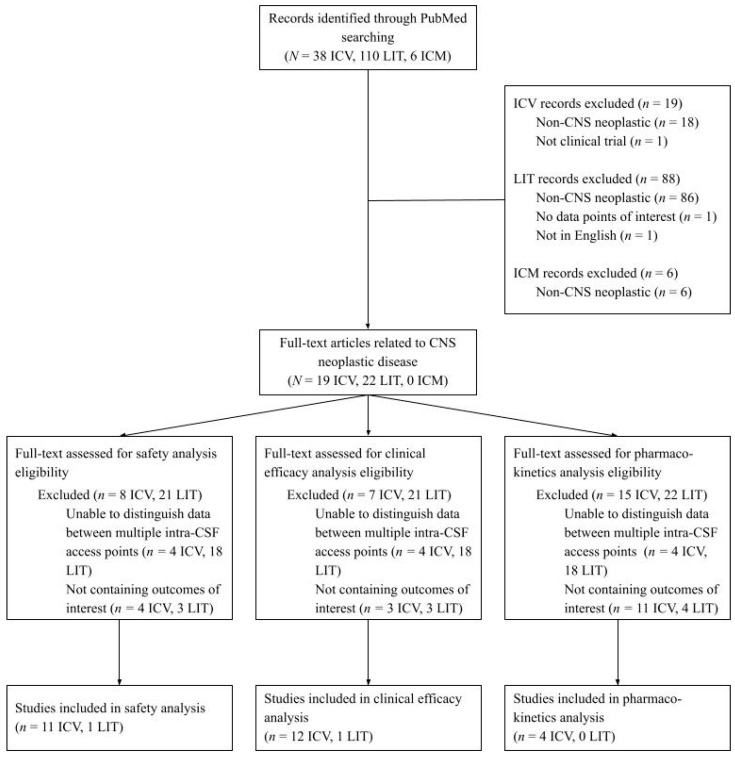
CONSORT diagram of the literature review process.

**Table 1 cancers-17-01263-t001:** (Adapted from Sadekar et al. [9]): Summary of ICV, LIT, and ICM drug delivery.

Delivery Route	Uses [10,11]	Advantages [7,12]	Limitations [9]	History
ICV	-Oncology -Pain management -Seizure/epilepsy -Neurodegenerative diseases -Infectious meningitis	-Widespread CNS delivery -Delivery at constant rate to minimize changes in intracranial pressure -Can be used for long-term administration -Minimizes systemic toxicity -Developed neurosurgery protocols	-Invasive surgery -Crosses parenchyma -Risk of neurosurgical complications -Risk of infectious complications	Ommaya 1963 [13]
LIT	-Oncology -Pain management -Spasticity -Neurodegenerative diseases -Infectious meningitis	-Minimally invasive -Routine outpatient procedure	-Longer distance to the brain -Influenced by posture -Require repeated punctures	Bier 1898 [14]
ICM	-Oncology -Pain management -Neurodegenerative diseases -Infectious meningitis	-Delivery closer to brain -Does not cross parenchyma	-Less developed surgical protocols -Risk of surgical complications	Ayer 1920 [15]

**Table 2 cancers-17-01263-t002:** ICV publications.

Author, Year	Disease (Tumor Type)	Study Phase	Patient Population (*N*)
Glitza Olivia et al., 2023 [18]	Leptomeningeal metastases (melanoma)	I/Ib	25
Kumthekar et al., 2022 [19]	Leptomeningeal metastases (breast cancer)	I/II	34
Li et al., 2023 [20]	Leptomeningeal metastases (lung cancer)	I	23
Sandberg et al., 2019 [21]	Recurrent ependymoma (posterior fossa)	Pilot	6
Mrugala et al., 2019 [22]	Leptomeningeal metastases (breast cancer)	II	3
Kramer et al., 2018 [23]	Medulloblastoma	II	43
Rubenstein et al., 2013 [24]	Recurrent CNS lymphoma (non-Hodgkin lymphoma)	I	14
Blaney et al., 2013 [25]	Pediatric neoplastic meningitis (leukemia/lymphoma or solid CNS tumor)	I	19
Glantz et al., 2010 [26]	Neoplastic meningitis (solid tumors of different origin)	Retrospective of phase IV	100
Groves et al., 2008 [27]	Meningeal malignancies (leukemia/lymphoma and solid tumors)	II	62
Slavc et al., 2003 [28]	Disseminated brain malignant tumors	Retrospective	26
Fleischhack et al., 2001 [29]	Brain metastases (medulloblastoma, primitive neuroectodermal tumor, glioblastoma, ependymoma)	Pilot	14

**Table 3 cancers-17-01263-t003:** LIT publications.

Author, Year	Disease	Drug	Study Phase	Patient Population (N)
Fan et al., 2021 [30]	Leptomeningeal metastases (EGFR-mutant NSCLC)	Pemetrexed combined with dexamethasone	I/II	30
Glantz et al., 2010 [26]	Neoplastic meningitis (NSCLC, primary CNS tumor, breast cancer)	sustained-release cytarabine or methotrexate	Retrospective of phase IV	100

**Table 4 cancers-17-01263-t004:** Safety in ICV delivery.

Author, Year	Grade	Possibly Related/Related AEs (Frequency, if Reported)
Glitza Olivia et al., 2023 [18]	1	Nausea (*n* = 7, 28%), dizziness (*n* = 4, 16%), vomiting (*n* = 3, 12%), paresthesia (*n* = 2, 8%), pruritis (*n* = 1, 4%), anorexia (*n* = 1, 4%), eye disorders (*n* = 1, 4%)
	2	Neck pain (*n* = 2, 8%), transient aphasia (*n* = 1, 4%)
Kumthekar et al., 2022 [19]	1	Headache (*n* = 3, 12%), noninfectious meningitis/arachnoiditis (*n* = 1, 4%), fatigue (*n* = 1, 4%), fever (*n* = 1, 4%), nausea (*n* = 1, 4%), malaise (*n* = 1, 4%), vertigo (*n* = 1, 4%), anorexia (*n* = 1, 4%)
	2	Noninfectious meningitis/arachnoiditis (*n* = 3, 12%), headache (n = 2, 8%), fatigue (*n* = 1, 4%), laryngitis (*n* = 1, 4%), vomiting (*n* = 1, 4%), back pain (*n* = 1, 4%), extremity pain (*n* = 1, 4%)
	3	Hydrocephalus (*n* = 1, 4%), nausea (*n* = 1, 4%)
	4	Noninfectious meningitis/arachnoiditis (*n* = 2, 8%)
Li et al., 2023 [20]	1	Elevation of ALT/AST (*n* = 5, 22%), myelosuppression (*n* = 2, 9%), anemia (*n* = 1, 4%)
	2	Anemia (*n* = 3, 13%), myelosuppression (*n* = 2, 9%), epilepsy (*n* = 1, 4%), scalp infection (*n* = 1, 4%)
	3	Myelosuppression (*n* = 3, 13%), epilepsy (*n* = 1, 4%), elevation of ALT/AST (*n* = 1, 4%)
	4	Myelosuppression (*n* = 1, 4%)
Sandberg et al., 2019 [21]	1	Vomiting (*n* = 3, 50%), nausea (*n* = 2, 33%), headache (*n* = 1, 17%), stomach cramps (*n* = 1, 17%)
	3	Reservoir infection (*n* = 1, 17%)
Mrugala et al., 2019 [22]	3	Transaminitis (*n* = 3, 100%)
	4	Lymphopenia (*n* = 1, 33%)
Kramer et al., 2018 [23]	2/3	Fever, headache, nausea, vomiting
	3	Transient acute bradycardia with somnolence (*n* = 2, 5%), headache, fatigue, pleocytosis, acute dystonic reaction
Rubenstein et al., 2013 [24]	1	Paresthesias, chills, rigors
	3/4	Lymphopenia (*n* = 2, 14%), fatigue, cataract, gait/CN III neuropathy, neutropenia, muscle weakness
Blaney et al., 2013 [25]	1	Electrolyte imbalance (*n* = 15, 79%), vomiting (*n* = 5, 26%), fatigue (*n* = 5, 26%), fever (*n* = 2, 11%), diarrhea (*n* = 2, 11%). nausea (*n* = 1, 5%), anorexia (*n* = 1, 5%), headache (*n* = 1, 5%), hepatic test abnormalities (*n* = 1, 5%), vision-blurred (*n* = 1, 5%)
	2	Arachnoiditis (*n* = 2, 11%), headache (*n* = 2, 11%), albumin abnormalities (*n* = 1, 5%), alopecia (*n* = 1, 5%), electrolyte imbalance (*n* = 1, 5%), hepatic test abnormalities (*n* = 1, 5%), vomiting (*n* = 1, 5%)
	3	Arachnoiditis (*n* = 2, 11%), electrolyte imbalance (*n* = 1, 5%), hepatic test abnormalities (*n* = 1, 5%), infection/febrile neutropenia (*n* = 1, 5%), nausea (*n* = 1, 5%)
	4	Headache (n = 1, 5%)
Groves et al., 2008 [27]	1/2	Chemical meningitis (*n* = 17, 65%), fatigue (*n* = 2, 8%), nausea or vomiting (*n* = 1, 4%), dyspnea (*n* = 1, 4%)
	3/4	CNS symptoms (*n* = 11, 42%), leukopenia (*n* = 4, 15%), constipation (*n* = 4, 15%), chemical meningitis (*n* = 3, 12%), anorexia (*n* = 3, 12%), nausea or vomiting (*n* = 3, 12%), dyspnea (*n* = 3, 12%), infection (*n* = 3, 12%), pain (*n* = 3, 12%), fatigue (*n* = 2, 8%), anemia (*n* = 2, 8%), hyponatremia (*n* = 2, 8%), thrombocytopenia (*n* = 1, 4%), chest pain (*n* = 1, 4%), diarrhea (*n* = 1, 4%), fever (*n* = 1, 4%), pruritus (*n* = 1, 4%), seizure (*n* = 1, 4%), upper GI bleed (*n* = 1, 4%), thrombosis (*n* = 1, 4%)
Slavc et al., 2003 [28]	N/A	Headache, nausea, neck pain, vomiting
Fleischhack et al., 2001 [29]	N/A	Headache, infection (meningitis) (2 of 59 courses), reservoir malfunction (*n* = 1, 7%), vomiting, temporary confusion, transient coma, generalized seizure associated with hyponatremia

Grading system for adverse events (AEs): Grade 1 are mild and generally not bothersome. Grade 2 events are bothersome and may interfere with doing some activities but are not dangerous. Grade 3 events are serious and interfere with a person’s ability to do basic things like eat or get dressed. Grade 3 events may also require medical intervention. Grade 4 events are usually severe enough to require hospitalization. Grade 5 events are fatal.

**Table 5 cancers-17-01263-t005:** Clinical efficacy in ICV delivery.

Author, Year	Patient Population (*N*) (Disease)	Drug Treatment	Response	Survival
Glitza Olivia et al., 2023 [18]	25 (melanoma)	Nivolumab	-	Median OS: 4.9 mo
Kumthekar et al., 2022 [19]	26 (breast cancer)	Trastuzumab	13 SD, 5 PR, 8 PD	Median PFS: 2.2 mo Median OS: 8.3 mo
Li et al., 2023 [20]	23 (lung cancer)	Pemetrexed	9 SD, 10 PR, 4 SD	Median PFS: 6.3 mo Median OS: 9.5 mo
Sandberg et al., 2019 [21]	6 (ependymoma)	5-Azacytidine	5 PD, 1 discontinued	-
Mrugala et al., 2019 [22]	3 (breast cancer)	Methotrexate and Liposomal Cytarabine	3 PD	Median PFS: 1.4 mo Median OS: 8.2 mo
Kramer et al., 2018 [23]	42 (medulloblastoma)	Radioimmunotherapy 131I-3F8	9 SD, 1 PR, 12 PD 15 CR, 5 PD	Median PFS: 11 mo
Rubenstein et al., 2013 [24]	14 (non-Hodgkin lymphoma)	Rituximab (1st treatment each week), rituximab + methotrexate (2nd treatment each week)	6 CR, 1 PR, 1 SD, 6 PD	-
Blaney et al., 2013 [25]	19 (leukemia/lymphoma or solid CNS tumor)	Topotecan	0 CR, 3 SD	-
Glantz et al., 2010 [26]	16 (NSCLC, primary CNS tumor, breast cancer)	liposomal cytarabine or methotrexate	-	PFS (ICV cytarabine): 43 days PFS (ICV methotrexate): 43 days
Groves et al., 2008 [27]	62 (leukemia/lymphoma and solid tumors)	Topotecan	18 SD, 10 PR, 12 PD	Median survival: 15 weeks
Slavc et al., 2003 [28]	11 alive	Mafosfamide and Etoposide	6 CR, 5 PR	-
Fleischhack et al., 2001 [29]	14 (medulloblastoma, primitive neuroectodermal tumor, glioblastoma, ependymoma)	Etoposide	5 PR, 3 PD, 6 SD	-

CR: complete response, PD: progressive disease, PR: partial response, SD: stable disease, OS: overall survival, PFS: progression-free survival.

**Table 6 cancers-17-01263-t006:** Pharmacokinetic studies with ICV delivery.

Author, Year	Drug Treatment	Dose	Patient Number	Results
Blaney et al., 2013 [22]	Topotecan	0.1 and 0.2 mg	18	Therapeutic target concentration of 1 ng/mL was reached in all patients at the 0.2 mg dose level.
Fleischhack et al., 2001 [29]	Etoposide	0.5 mg	4	The terminal half-life in the CSF was 7.4 ± 1.2 h, and the area under the curve was 25.0 ± 9.5 μg·h/mL. Volume of distribution at steady state averaging 0.16 L and total clearance averaging 0.46 mL/min.
Rubenstein et al., 2013 [24]	Rituximab and methotrexate	10 mg or 25 mg Rituximab and 12 mg methotrexate	14	Biphasic decline in CSF rituximab concentrations, peak levels at 580 μg/mL at the 25 mg dose. The elimination rate of rituximab was slower when co-administered with methotrexate (0.36/h) compared to rituximab monotherapy (0.84/h).
Kumthekar et al., 2022 [19]	Trastuzumab	80 mg	10	Mean volume of distribution of 73 ± 48 mL and a clearance rate of 14 ± 5 mL/h. The apparent CSF half-life was relatively short at 4.1 ± 3.0 h.

**Table 7 cancers-17-01263-t007:** Safety in LIT delivery.

Author, Year	Grade	Possibly Related/Related AEs (Frequency, If Reported)
Fan et al., 2021 [30]	1	Vomiting (*n* = 6, 20%), nausea (*n* = 2, 7%), limb pain (*n* = 1, 3%), back pain (*n* = 1, 3%)
	2	Myelosuppression (*n* = 6, 20%), limb pain (*n* = 2, 7%), paralysis (*n* = 2, 7%), headache (*n* = 2, 7%), back pain (*n* = 1, 3%)
	3	Myelosuppression (*n* = 4, 13%), limb pain (*n* = 2, 7%), headache (*n* = 1, 3%)
	4	Myelosuppression (*n* = 1, 3%)

## Data Availability

The original contributions presented in this study are included in the article. Further inquiries can be directed to the corresponding author.

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
