# Peer review of "Retrospective Review of Intra-Cerebrospinal Fluid (CSF) Drug Delivery in CNS Malignancies: Safety, Clinical Efficacy and Pharmacokinetic Profiles of Intracerebroventricular (ICV), Lumbar Intrathecal (LIT), and Intra-Cisterna Magna (ICM) Injections"

_cancers, 2025, doi:10.3390/cancers17081263_

Round 1

Reviewer 1 Report

Comments and Suggestions for Authors

A useful summary of the literature to date of methods of CSF drug delivery. 

Tackles a difficult area with little evidence and provides a platform for further research considerations.

I've no specific changes to recommend 

Author Response

We thank Reviewer 1 for their time and review of our manuscript.

Reviewer 2 Report

Comments and Suggestions for Authors

This is an interesting review paper about intra-CSF drug delivery, comparing among ICV, LIT, and ICM. This paper may be worthy of attention to tackle CNS malignancies in the future. I have a few comments about it.

#Introduction

It says that ICV, LIT, and ICM are the three main access points for intra-CSF delivery. I understand that these are the main three, but how about others? I do not know what else is available for access points for intra-CSF delivery; but ICV, LIT, and ICM must have, as a whole, some more advantages than others. My question is:

-Are there any access points for intra-CSF delivery other than ICV, LIT, and ICM?

-If so, have these others been already discouraged? What is the advantage, as a whole, of the main three? Since these are the three that are the main access points, there must be something more favorable for them.

-Considering this way, however, ICM is already the discouraged one, isn’t it? In the present MS, it seems that ICM is being discussed on a par with ICV or LIT. Please re-consider a little how ICM should be dealt with/discussed in this paper.

#Table 1

This table compared the factors, including Uses, Advantages, and Limitations, of the three modalities. The other aspects I want to know more among the three are:

-Cost: What is the cost performance? How much it costs is an important factor to effectively implement it in the clinical medicine.

-History: When was each of the three modalities implemented in the clinical medicine? An appropriate reference, for example, the first paper about the modality, may be informative and interesting.

#Table 2

In this table, there are terms like ‘metastasis’, ‘neoplastic meningitis’, or ‘meningeal malignancies.’ But is it better to show, for example, ‘metastasis of what tumor’? Is it a metastasis of lung cancer? Are they meningeal malignancies of what? Are they meningeal malignancies due to lymphomatous involvement or something else? If possible, a certain amount of information would be nice to be included in this table. 

#Table 3

-The information of drugs delivered may be necessary. This is because the AEs are not solely owing to the delivery modality of ICV itself, but also to the kind of drugs delivered. For instance, chemotherapeutic agents must have caused more symptoms like nausea or vomiting than others.

-‘Grade’ must be something of some scale for AEs, right? But most readers, including this reviewer, do not know. Each of the grades should be shown or described/tabulated and/or at least the scale should be cited with appropriate references.

#Table 4

This is pertinent to the above. It would better if this table had the information of the diseases targeted and/or the drugs administered. This is because the efficacy (Response/Survival) is not solely determined by the delivery system of ICV itself, but also by these two factors, probably.

#3.1.3 Pharmacokinetics

This section is full of sentences. They are understandable, but I just wonder if it is possible to make a table as well for this section for better understanding. If possible, please consider it.

#3.2. Lumbar intrathecal findings

For LIT as well, the sections or tables corresponding to those of ICV are given in here. I hope the authors make the changes as well for these sections or tables just similarly to those of ICV.

#Conclsions

This section may be important. But should it be really separated from Discussion? Maybe, it can be incorporated into the Discussion.

Author Response

This is an interesting review paper about intra-CSF drug delivery, comparing among ICV, LIT, and ICM. This paper may be worthy of attention to tackle CNS malignancies in the future. I have a few comments about it.

#Introduction
It says that ICV, LIT, and ICM are the three main access points for intra-CSF delivery. I understand that these are the main three, but how about others? I do not know what else is available for access points for intra-CSF delivery; but ICV, LIT, and ICM must have, as a whole, some more advantages than others. My question is: Are there any access points for intra-CSF delivery other than ICV, LIT, and ICM?

No, there are no additional access points for CSF delivery and generally speaking most only consider ICV and LIT the only two methods, but because there are a few very rare instances of ICM we have included it.

If so, have these others been already discouraged? What is the advantage, as a whole, of the main three? Since these are the three that are the main access points, there must be something more favorable for them.

There are no others that have been excluded. And yes, there are pros and cons as we have outlined herein.

Considering this way, however, ICM is already the discouraged one, isn’t it? In the present MS, it seems that ICM is being discussed on a par with ICV or LIT. Please re-consider a little how ICM should be dealt with/discussed in this paper.

Correct, ICM is not typically used. We included it so that we are comprehensive in our reporting of the data. We thank reviewer #2 for these questions. We have added the following to the text on page 3 in order to address this comment: “Decision on which CNS delivery location is most suitable should be made based on the targeted area(s). For instance, the lumbar IT approach is preferred for spinal cord injury as it enables high local concentrations in the spinal cord. In contrast, one might choose ICM to achieve higher concentrations, especially in the brain stem but the feasibility of this route in
humans is questionable. The cisterna magna is a large CSF compartment and is frequentlyused in studies with non-human primates and other animals, it is infrequently used as an access point in humans mainly because the idea of inserting a needle near the vital centers of the medulla oblongata could cause brain stem injury.”

#Table 1
This table compared the factors, including Uses, Advantages, and Limitations, of the three modalities. The other aspects I want to know more among the three are:

We thank the reviewer for bringing up this very interesting point on cost. There are many variables that impact patient cost including location of treatment (outpatient clinic setting vs inpatient), drug given, who is delivering drug (physician), cost of drug itself and whether or not CSF sampling is being done. We have added these comments within the manuscript on page 4.

History: When was each of the three modalities implemented in the clinical medicine? An appropriate reference, for example, the first paper about the modality, may be informative and interesting.

We thank Reviewer 2 for this comment and agree that it would be of interest. As such, we have modified Table 1 with added history column that addresses this and we have also added the below references respectively:
Ayer JB., Puncture of the cisterna magna, Arch Neurol Psychiatr, 1920; 4: 529–41
Ommaya A. Subcutaneous reservoir and pump for sterile access to ventricular cerebrospinal fluid. Lancet. 1963; 282: 983–4.
Bier, A (1899). "Versuche uber cocainisirung des ruckenmarkes (Experiments on the cocainization of the spinal cord)". Deutsche Zeitschrift für Chirurgie (in German). 51 (3–4): 361–9.

#Table 2
In this table, there are terms like ‘metastasis’, ‘neoplastic meningitis’, or ‘meningeal malignancies.’ But is it better to show, for example, ‘metastasis of what tumor’? Is it a metastasis of lung cancer? Are they meningeal malignancies of what? Are they meningeal malignancies due to lymphomatous involvement or something else? If possible, a certain amount of information would be nice to be included in this table.

We thank Reviewer 2 for pointing this out and we have added clarification on tumor type accordingly.

#Table 3
The information of drugs delivered may be necessary. This is because the AEs are not solely owing to the delivery modality of ICV itself, but also to the kind of drugs delivered. For instance, chemotherapeutic agents must have caused more symptoms like nausea or vomiting than others.

We agree with Reviewer 2 and added this point accordingly in the manuscript on p 7.

‘Grade’ must be something of some scale for AEs, right? But most readers, including this reviewer, do not know. Each of the grades should be shown or described/tabulated and/or at least the scale should be cited with appropriate references.

Thank you for this comment, clarification of the grading system added on page 9 of the manuscript.

#Table 4
This is pertinent to the above. It would better if this table had the information of the diseases targeted and/or the drugs administered. This is because the efficacy (Response/Survival) is not solely determined by the delivery system of ICV itself, but also by these two factors, probably.

We thank Reviewer 2 for this comment and as such drug and disease added to the table accordingly.

#3.1.3 Pharmacokinetics
This section is full of sentences. They are understandable, but I just wonder if it is possible to make a table as well for this section for better understanding. If possible, please consider it.

We appreciate this comment and as a result we have added Table 5 on page 10 accordingly.

#3.2. Lumbar intrathecal findings

For LIT as well, the sections or tables corresponding to those of ICV are given in here. I hope the authors make the changes as well for these sections or tables just similarly to those of ICV.
Thank you for this comment, we have updated table 6 with tumor type and drug accordingly.

#Conclsions
This section may be important. But should it be really separated from Discussion? Maybe, it can
be incorporated into the Discussion.

Per Reviewer 2 request, we have combined the conclusion
within the discussion section.

Reviewer 3 Report

Comments and Suggestions for Authors

In this work, Lee and colleagues present a retrospective literature review of existing reports of intra-cerebrospinal fluid (CSF) drug delivery for the treatment of CNS malignancies. The goal of this review was to compare the safety, efficacy, and pharmacokinetic profiles of three intra-CSF delivery methods: intracerebroventricular (ICV), lumbar intrathecal (LIT), 20 and intra-cisterna magna (ICM). After a thorough literature search, they identified a total of 11 reports which met criteria for inclusion in this study. Overall, clinical outcomes and safety data varied across these studies, making it difficult to draw broad conclusions about this therapeutic approach, also highlighting the need for further study of these techniques to determine their efficacy and potential role in CNS tumor treatment.  This report is well written and is a good foundation for further inquiry into this important therapeutic approach.

Author Response

In this work, Lee and colleagues present a retrospective literature review of existing reports of intra-cerebrospinal fluid (CSF) drug delivery for the treatment of CNS malignancies. The goal of this review was to compare the safety, efficacy, and pharmacokinetic profiles of three intra-CSF delivery methods: intracerebroventricular (ICV), lumbar intrathecal (LIT), 20 and intra-cisterna magna (ICM). After a thorough literature search, they identified a total of 11 reports which met criteria for inclusion in this study. Overall, clinical outcomes and safety data varied across these studies, making it difficult to draw broad conclusions about this therapeutic approach, also highlighting the need for further study of these techniques to determine their efficacy and potential role in CNS tumor treatment. This report is well written and is a good foundation for further inquiry into this important therapeutic approach.

We thank Reviewer 3 for their time and review of our manuscript.

Reviewer 4 Report

Comments and Suggestions for Authors

the authors provide a nice review of intrathecal methods of drug delivery. This article does not seem to be a systematic review, but nonetheless covers significant aspects of the literature. pharmacokinetics data can hardly be generalized since heavily dependent on patient population and infusate - should discuss. Similarly, should briefly mention the role of surgical navigation for Ommaya implantation, which has reduced complications.

Author Response

the authors provide a nice review of intrathecal methods of drug delivery. This article does not seem to be a systematic review but nonetheless covers significant aspects of the literature. pharmacokinetics data can hardly be generalized since heavily dependent on patient population and infusate - should discuss. Similarly, should briefly mention the role of surgical navigation for Ommaya implantation, which has reduced complications. We thank Reviewer 4 for their time and review of our manuscript.

We have added comments within the discussion on p12 to address this potential heterogeneity in pharmacokinetics data.

Round 2

Reviewer 2 Report

Comments and Suggestions for Authors

Thank you for the revision. I am sure the revised MS is much better than the original one.